# Experimental and Numerical Investigation Integrated with Machine Learning (ML) for the Prediction Strategy of DP590/CFRP Composite Laminates

**DOI:** 10.3390/polym16111589

**Published:** 2024-06-03

**Authors:** Haichao Hu, Qiang Wei, Tianao Wang, Quanjin Ma, Peng Jin, Shupeng Pan, Fengqi Li, Shuxin Wang, Yuxuan Yang, Yan Li

**Affiliations:** 1School of Materials Science and Engineering, Hebei University of Technology, Tianjin 300401, China; 2School of Mechanical and Engineering, Tianjin Sino-German University of Applied Sciences, Tianjin 300350, China; 15631688218@163.com (T.W.); jinpeng202401@163.com (P.J.); fengyinghezuo@foxmail.com (S.P.); fengqi18074837032@163.com (F.L.); a17534929808@163.com (S.W.); yyx19834083198@163.com (Y.Y.); 3School of Mechanical and Engineering, Hebei University of Technology, Tianjin 300401, China; 4School of System Design and Intelligent Manufacturing, Southern University of Science and Technology, Shenzhen 518055, China; maqj@sustech.edu.cn; 5Tianjin Sino-Spanish Machining Tool Vocational Training Center, Tianjin Sino-German University of Applied Sciences, Tianjin 300350, China; liyan@tsguas.edu.cn

**Keywords:** machine learning, stacking sequence, orientation selections, mechanical properties, finite element modeling, prediction strategy

## Abstract

This study unveils a machine learning (ML)-assisted framework designed to optimize the stacking sequence and orientation of carbon fiber-reinforced polymer (CFRP)/metal composite laminates, aiming to enhance their mechanical properties under quasi-static loading conditions. This work pioneers the expansion of initial datasets for ML analysis in the field by uniquely integrating the experimental results with finite element simulations. Nine ML models, including XGBoost and gradient boosting, were assessed for their precision in predicting tensile and bending strengths. The findings reveal that the XGBoost and gradient boosting models excel in tensile strength prediction due to their low error rates and high interpretability. In contrast, the decision trees, K-nearest neighbors (KNN), and random forest models show the highest accuracy in bending strength predictions. Tree-based models demonstrated exceptional performance across various metrics, notably for CFRP/DP590 laminates. Additionally, this study investigates the impact of layup sequences on mechanical properties, employing an innovative combination of ML, numerical, and experimental approaches. The novelty of this study lies in the first-time application of these ML models to the performance optimization of CFRP/metal composites and in providing a novel perspective through the comprehensive integration of experimental, numerical, and ML methods for composite material design and performance prediction.

## 1. Introduction

With the growing emphasis on environmental protection and energy scarcity, the industrial demand for lightweight, high-strength materials, especially reinforced composite materials, has rapidly increased [1]. Despite the high performance of carbon fiber composite materials, they still exhibit some disadvantages compared to traditional metal materials, such as poor impact resistance, high cost, and low flexibility. To overcome these issues, fiber/metal composite laminate materials have emerged [2,3,4,5]. This material combines the advantages of metal and fiber composite materials, exhibiting high strength and stiffness while maintaining relatively low density, excellent fatigue performance, and high damage tolerance. It has demonstrated superior performance in impact resistance, energy absorption, vibration reduction, heat dissipation, and sound insulation [6,7,8,9]. The results have shown that steel/carbon fiber composite materials have a higher load-bearing capacity and better energy absorption performance than single-material structures, making them highly applicable in the automotive and rail transportation industries [10]. This combination offers structural strength and lightweight design advantages, providing innovative solutions for various engineering applications [11,12,13,14,15]. To fully harness the performance of these materials, a thorough understanding of their reliability during service is crucial. Tensile and bending are key indicators of their fundamental mechanical properties [16,17,18,19,20,21,22,23,24,25]. Tensile performance directly affects the stability and ductility of the material under tensile stress, forming the basis for evaluating its overall strength. Meanwhile, bending performance plays a crucial role in the practical application of composite structures, as many engineering scenarios involve bending rather than pure tension. By understanding the behavior of fiber/metal composite materials under tensile and bending loads, we can more accurately predict their performance in real-world applications, optimizing the design and ensuring their reliability and stability in various engineering applications.

In traditional materials performance evaluation, experimental testing has been an indispensable means; however, the time-consuming and costly nature of experiments has posed challenges for researchers [26,27,28]. The complexity of experiments and sample preparation requirements limits the speed and range of testing, especially for composite materials that need to consider multiple load conditions in predicting long-term performance. In this context, numerical simulation techniques have become one of the new approaches researchers have explored. Numerical simulation, by establishing mathematical models, simulates the behavior of materials under different environments and loading conditions, providing valuable insights for material design [13,23,29,30]. Despite the significant progress that numerical simulation methods have brought to materials research, they still face challenges related to massive computational resource requirements and high dependence on model accuracy. Pushkar et al. investigated ballistic penetration in laminated plates as complex but crucial for designing effective structural protection. Experimental means are expensive and often dangerous, while numerical simulation, though a good supplement, is computationally intensive. Developing and testing an efficient tool for the real-time prediction of projectile penetration through laminated plates has been addressed by training neural networks and decision tree regression models [31].

In recent years, with the vigorous development of machine learning technology, there have been new possibilities for predicting the performance of composite materials. Christian et al. summarized the latest developments using machine learning for enhanced composite material design in lightweight industrial structures. They discussed the limitations of traditional methods and provided detailed schemes for introducing machine learning into composite material technology, focusing on implementing machine learning algorithms, data cleaning, material/process selection, and data acquisition techniques. The assessment covered emerging digital tools platforms for implementing machine learning algorithms and pointed out research gaps in composite material design for the future [32]. The uniqueness of machine learning lies in its ability to learn complex patterns of material behavior from large amounts of data and predict its performance in untested conditions [33,34,35,36,37]. Some machine learning applications in composite material replace traditional experiments and numerical simulations and provide researchers with a new perspective. By learning from data, machine learning models can capture nonlinear relationships and complex patterns that are difficult to obtain through traditional means, providing profound insights. It is crucial to understand the performance variations, challenges, and advantages of composite materials under different load conditions.

Many researchers have used machine learning models to predict the performance of composite laminates under different load conditions. Faramarz et al. and Christos et al. predicted the non-destructive strength of composite laminate panels based on deep learning and stochastic finite element methods [38]. Chen et al. developed efficient data-driven models by implementing integrated learning algorithms such as “gradient boosting decision trees” and “random forests” on a collected database of steel/fiber single-lap shear tests. The results showed that the model generated by the gradient boosting decision tree algorithm achieved the best accuracy (R^2^ = 0.98) in predicting steel/fiber interface adhesion strength, surpassing other integrated and machine learning algorithms [36].

However, despite the enormous innovative potential that machine learning technology brings to composite material research, its application still has challenges. One major challenge is the quality and diversity of data. The performance of machine learning models heavily depends on the quality and diversity of their training data. Acquiring large-scale and high-quality data is an urgent issue that needs to be addressed. Chahar et al. pointed out that machine learning requires a substantial amount of high-fidelity data, advocating using multi-fidelity (MF) machine learning algorithms based on Gaussian processes. They suggested combining these algorithms with finite element simulations to quantify uncertainties through training surrogate models [39].

Additionally, model interpretability is a crucial issue, especially in highly complex composite material systems [36,40,41,42]. Machine learning has paved a new path for composite material research, providing more efficient and economical means to predict and understand material performance. With a deepening understanding of data and algorithms, coupled with continuous improvements in related technologies, machine learning is expected to play an increasingly important role in the future of composite material engineering, driving more innovations and breakthroughs [41,43,44,45,46].

This paper is dedicated to comprehensively studying the mechanical properties of metal/CFRP composite materials under tension and bending conditions through the integrated application of machine learning, numerical simulation, and experimental testing. Firstly, we established a performance database for the tension and bending of metal/CFRP composite materials through experimental tests. Subsequently, we developed finite element simulation models to simulate the tensile and bending behavior of metal/CFRP composite laminates. The model considered material nonlinearity, anisotropy, and interlayer bonding. The established finite element model could accurately predict the experimental data. The finite element simulation data, combined with the experimental data, formed a high-precision dataset for training machine learning models to predict the behavior of composite materials in untested conditions.

Furthermore, the prediction accuracy of different models is extensively discussed, as well as their applicability and limitations in various scenarios. The aim is to comprehensively assess material performance and deepen the understanding of material behavior through finite element simulation and machine learning methods. This integrated research approach can contribute to a more in-depth understanding of composite material performance and provide a scientific basis for future engineering design and material selection.

## 2. Machine Learning Methods

In the study of the mechanical performance of DP590/CFRP composite panels under tensile and bending conditions, a selection of nine machine-learning models was made to encompass a broad range of algorithmic families. This selection was guided by several criteria crucial for robust prediction capabilities in the context of material behavior. These criteria included the complexity and nonlinearity of the data patterns, the limited amount of data available for training, and the requirement for interpretability, which is essential in materials science.

Three major classes of algorithms—linear, nonlinear, and tree-based models—were included to leverage their distinct advantages under different data characteristics. Linear models, namely linear regression, ridge regression, and lasso regression, were selected for their efficiency and the provision of a baseline in scenarios where relationships might predominantly appear linear or when baseline model performance comparisons were necessary.

Nonlinear models, such as K-nearest neighbors and polynomial regression, were chosen for their capacity to model complex relationships beyond linear interactions, particularly valuable in datasets that exhibit nonlinear patterns, as anticipated in the behavior of composite materials.

Tree-based models including decision trees, random forests, gradient boosting, and XGBoost were incorporated due to their superior ability to manage complex, high-dimensional data structures, which are typical in materials engineering. These models excel in handling interactions and heterogeneities inherent in composite material data.

The choice to exclude more complex algorithms like deep learning stemmed from the inadequate volume of available data, which might lead to overfitting, and the necessity for computational efficiency and interpretability in results, which are critical in applied material research.

### 2.1. Introduction to Machine Learning Models

#### 2.1.1. Linear Regression

Linear regression was chosen for modeling simple relationships [47]. Due to its simplicity and interpretability, it is suitable for cases with an approximate linear relationship between features and the target. Linear regression is a benchmark that helps us understand the linear relationships between features.

The basic formula for linear regression is
(1)$$y=b_{0}+b_{1}x_{1}+b_{2}x_{2}+\ldots +b_{n}x_{n}$$
where y is the target variable, x_1_, x_2_, …, x_n_ are the feature variables, and b_0_, b_1_, b_2_, …, bn are the model parameters.

In the provided information, y represents the target variable, and x_1_, x_2_, …, x_n_ represent the feature variables, while b_0_, b_1_, b_2_, …, b_n_ denote the model parameters.

Linear regression is based on the least squares method, aiming to minimize the sum of the squared differences between the actual observed and model-predicted values. By solving for the coefficients b, the model obtains the best-fitting line for the data. This model assumes a linear relationship between the target variable and the features.

#### 2.1.2. Ridge Regression

Ridge regression is a regularized linear regression method applicable in multicollinear situations (high correlation among features) [48]. By introducing a regularization term, ridge regression helps to prevent overfitting, improving the model’s generalization ability. The chosen ridge regression addresses the possible correlation among the features.

The objective function of ridge regression is
(2)$$J\left(\beta \right)=\sum_{i=1}^{m}{\left(y_{i}-\beta_{0}-\sum_{j=1}^{n}\beta_{j}x_{ij}\right)}^{2}+\alpha \sum_{j=1}^{n}\beta_{j}^{2}$$
where J(β) is the loss function, β_1_, β_2_, …, β_n_ are the coefficients, and α is the regularization coefficient.

By introducing an L2 regularization term based on the least squares method, ridge regression aims to prevent model overfitting. The regularization term penalizes coefficients, causing the model’s coefficients to lean more towards zero, thereby alleviating the collinearity problem.

#### 2.1.3. Lasso Regression

Lasso regression is another type of regularized linear regression method, but unlike ridge regression, it uses L1 regularization. A characteristic of lasso regression is that it can produce sparse models, i.e., automatically selecting unimportant features [49]. In our study, choosing lasso regression helps discover the most important features for predicting composite material performance.
(3)$$J\left(\beta \right)=\sum_{i=1}^{m}{\left(y_{i}-\beta_{0}-\sum_{j=1}^{n}\beta_{j}x_{ij}\right)}^{2}+\alpha \sum_{j=1}^{n}\left|\beta_{j}\right|$$
where J(β) is the loss function, β_1_, β_2_, …, β_n_ are the coefficients, and α is the regularization coefficient.

Lasso regression also introduces a regularization term based on the least squares method but uses L1 regularization. It causes some coefficients to become zero, achieving the result of automatic feature selection, suitable for high-dimensional datasets.

#### 2.1.4. K-Nearest Neighbors

K-nearest neighbors (K-NN) is an instance-based learning method suitable for non-parametric models [50]. In our study, we chose K-NN to account for the local relationships between features, i.e., similar features may have similar mechanical properties. K-NN can flexibly adapt to the local structure of the data, and it performs well in modeling nonlinear relationships.
(4)$$\hat{y}=\frac{1}{k}\sum_{i=1}^{k}y_{i}$$
where $\hat{y}$ is the predicted value of the target variable, and y_i_ is the target value of the k nearest neighbors to the input sample.

K-nearest neighbors work based on a voting mechanism among neighbors. It predicts a target value through the average of the nearest target values. It is assumed that outputs with similar attributes are likely to have similar results, making them suitable for strong local relationships.

#### 2.1.5. Polynomial Regression

Polynomial regression is used to handle the nonlinear relationship between features and targets [51]. By introducing higher-order terms of features, polynomial regression can more flexibly adapt to the nonlinear structure of the data. In our study, considering that the performance of composite materials may be subject to complex nonlinear effects, we chose polynomial regression to capture these relationships better.
(5)$$y=b_{0}+b_{1}x+b_{2}x^{2}+\ldots +b_{n}x^{n}$$

In this context, y is the target variable, x refers to the feature variables, and b_0_, b_1_, b_2_, …, b_n_ are the model parameters. These parameters are estimated during the training process to minimize the difference between the predicted and the actual values of the target variable.

By adding higher-order terms of the features, polynomial regression allows the model to adapt to the nonlinear relationships within the data. Essentially, it is an extension of linear regression.

#### 2.1.6. Decision Tree

A decision tree is a tree-shaped model suitable for handling complex nonlinear relationships and interactive effects [52]. Decision trees divide the data space recursively, capturing the nonlinear relationships between features. In our study, we chose decision trees to consider possible nonlinear structures and provide interpretability through the structure of the tree. The prediction formula of a decision tree is a tree structure where each node represents a feature, each branch represents a decision rule, and the leaf nodes contain the prediction value of the target variable. Decision trees are based on recursive divisions of features, and they choose features that increase data purity. They are suitable for nonlinear relations and interactive effects and enhance interpretability.

#### 2.1.7. Random Forest

Random forest is an ensemble learning method based on decision trees, which improves the robustness and accuracy of the model by combining the predictions of multiple decision trees [53]. In our study, we chose random forest to harness the advantages of multiple decision trees and adapt to more complex model structures.

Random forest consists of an ensemble of decision trees, and the prediction result is the average or voting result of all trees. Random forest improves the robustness and accuracy of the model by training multiple decision trees and aggregating their predictions. Each tree is trained on a different subset of data, introducing randomness.

#### 2.1.8. Gradient Boosting

Gradient boosting is an ensemble learning method that trains weak learners iteratively and combines them into a strong learner [54]. Gradient boosting was chosen to enhance the model’s predictive performance further. Gradient boosting has a strong ability to fit complex nonlinear relationships.
(6)$$F\left(x\right)=\sum_{m=1}^{M}\gamma_{m}h_{m}(x)$$

In this context, F(x) is the final predicted result, γ_m_ represents the weight of the weak learner, and hm(x) is the predicted result of the weak learner. This method aims to combine multiple vulnerable learners to form a strong model.

Gradient boosting works by iteratively training multiple weak learners, each time adjusting the model to reduce the gradient of the loss function. It is suitable for nonlinear relationships and can gradually improve the performance of the model.

#### 2.1.9. XGBoost

XGBoost is an improved version of the gradient boosting algorithm, providing superior efficiency and performance [54]. XGBoost regression was selected to take full advantage of its exceptional training speed and superior generalization performance.
(7)$$F\left(x\right)=\sum_{m=1}^{M}\gamma_{m}h_{m}(x)+\sum_{k=1}^{K}\Omega (f_{k})$$

The term Ω(fk) is the regularization term for each tree in the model. This term is used to constrain the complexity of the tree structure and prevent overfitting.

XGBoost, building upon gradient boosting, introduces a regularization term to optimize model complexity, enhancing the training speed and performance. In this complex machine learning task, XGBoost offers considerable advantages, including high predictive performance, robustness against features, and the capability to handle missing values. These advantages make it a powerful modeling tool for predicting the mechanical properties of DP590/CFRP composite laminate.

### 2.2. Fine-Tuning Machine Learning Models

To enhance the model’s performance, we employed two tuning methods: grid search and cross-validation. Grid search is a tuning technique to find the optimal hyperparameters for the model. It works by specifying a grid of potential values for the hyperparameters and then systematically working through multiple combinations of those hyperparameters. Cross-validation evaluates these models to determine which combination offers the highest validation score. The model’s performance would thereby be significantly enhanced with the optimal parameters.

Cross-validation is a resampling procedure used to evaluate the model’s performance on a limited data sample. The most common cross-validation method is k-fold cross-validation, where ‘k’ is the number of groups a dataset sample will split into.

After separating the data into k groups, we fit the model using k-1 groups and evaluate the model’s performance on the remaining part of the data. We carry out this process k times, so we obtain k models and performance estimates, which we then average, providing a more accurate model performance indicative of the model’s ability to generalize to unseen data. By using systematic parameter search and validation, we aim to find the best model parameters to ensure the model’s robust generalization ability.

### 2.3. Machine Learning Model Evaluation Indicators

It will employ several evaluation metrics, including MAE (mean absolute error), MSE (mean squared error), R^2^ (coefficient of determination), and MAPE (mean absolute percentage error). These metrics will comprehensively assess the model’s performance in various aspects, ensuring comprehensive and accurate predictions of the composite material’s properties. The MAE measures the average magnitude of the errors in a set of predictions without considering their direction. It is the average of the absolute differences between prediction and actual observation over the test sample, where all individual differences have equal weight. The MSE, on the other hand, measures the average squared difference between the estimated and actual values. It is more sensitive to outliers than the MAE, as the differences are squared before they are averaged. The R^2^ or coefficient of determination is a statistical measure representing the proportion of the variance for a dependent variable explained by an independent variable or variables in a regression model. The closer to 1 this value is, the better the regression line fits. Lastly, the MAPE is often used for time series forecasting as it gives a simple percentage error that describes the average error rate for the forecast. The smaller the MAPE value, the better the accuracy of the forecast. By utilizing all these evaluation metrics, we can have a much more holistic view of our model’s performance, acknowledging its strengths and areas where it can improve.

## 3. Machine Learning Data Acquisition

### 3.1. Experimental Introduction

The process involves awakening the prepreg for about 0.5 h, then cutting and laying up, including placing a steel plate, molding, and finally, demolding and cooling. The stacking sequence of CFRP layers in the ply design of the DP590/CFRP composite laminates for both the tension and bending models is shown in Figure 1. The tensile strength test was conducted in accordance with the specifications of the ASTM-D638 standard [55], while the flexural strength test was carried out following the ASTM-D790 standard [56].

### 3.2. Introduction to Finite Element Modeling

The numerical model implemented a three-dimensional failure model based on strain damage law using the explicit finite element subroutine Abaqus-VUMAT. The interface failure was simulated using a bilinear cohesive force contact model, and the steel plate layer was described by the Johnson–Cook model. For the fiber layer, the 2D Hashin criterion, which only considers the in-plane stress components, was replaced by a 3D Hashin failure criterion developed based on VUMAT. In the program design, the initiation of fiber failure and matrix tensile failure is controlled by the maximum stress criterion and the strain-based Hashin criterion. In contrast, matrix compression failure is assessed using the Puck failure criterion.

The established finite element model can accurately predict the response and failure mechanisms of composite laminates during tensile and bending processes. The numerical model accurately predicted the experimental curves, notably achieving a 97% accuracy in predicting the maximum load. In the bending tests, both the experimental and numerical models demonstrated typical bilinear behaviors, with prediction accuracies reaching 88% for the bending modulus and 97% for the bending strength. Finally, utilizing the established FEA-VUMAT model, we studied the ultimate failure modes within and between layers. It was found that the predicted failure modes in the bending tests were consistent with the experimental results [57].

### 3.3. Ply Stacking Design of CFRP Layers in DP590/CFRP Composite Laminates

Utilizing experimentally validated finite element models, we extensively discussed the impact of different ply stacking sequences on the strength of DP590/CFRP composite laminates under tension and bending loading conditions. The analysis encompasses the reasons for strength variations, interlaminar stress distributions, and other aspects, aiming to understand the performance characteristics comprehensively.

In this study, 28 different layup sequences were selected to explore the mechanical properties of composite laminates, incorporating commonly used fiber angles such as 0°, ±45°, and 90°. The design included not only symmetric layups but also asymmetric configurations. This approach aims to comprehensively assess the impact of these varying fiber angles on the performance of the composite laminates. By integrating both symmetric and asymmetric layups, this study not only investigates the balance and stability of symmetric layups but also evaluates the performance advantages of asymmetric layups in specific applications. The specific layup sequences are presented in Table 1.

### 3.4. Data Augmentation and Preprocessing

This paper obtains tensile and bending strength data for multiple sets of DP590/CFRP composite laminates through experimental means. Subsequently, the finite element model is validated using the experimental data. To establish the finite element modeling, the tensile and bending strengths of composite laminates with different CFRP layup sequences were further analyzed. The initial numerical values obtained through the finite element and experimental methods amounted to 48 sets. The size of the training set was expanded through data augmentation methods from the original 48 sample sets to 1296 sets. Feature encoding was performed by applying one-hot encoding to the angle values in the “layer sequence”, and the encoded angle values were merged with the original data to handle angle information better. Standardization was employed to eliminate the dimensional differences between different features, scaling all features to a range with a mean of 0 and a standard deviation of 1. It helps in improving the training effectiveness and stability of the model. Furthermore, correlation analysis was conducted, and the correlation coefficients between features and strength were calculated. Only features with non-zero standard deviations were considered, with invariant features being excluded to minimize their impact on the results. The results of the correlation analysis are presented in the form of heatmaps, visually illustrating the degree of correlation between various features. The original data and expanded experimental data used in this study are available for download in the Appendix A section at the end of this article.

Figure 2, Figure 3, Figure 4 and Figure 5 compare the correlation coefficient heatmaps of the tensile strength and bending strength for DP590/CFRP composite plates before and after data augmentation. In these heatmaps, the *X*-axis and *Y*-axis represent different variables or parameters used in this study, including the material’s angles at various stacking sequences, roughness, cross-sectional thickness, cross-sectional width, cross-sectional area, and tensile strength. Each cell in the heatmap represents the correlation coefficient between the corresponding variables, ranging from −1 to 1, where 1 indicates a perfect positive correlation, −1 indicates a perfect negative correlation, and 0 indicates no correlation. Warm colors indicate positive correlations, meaning that as one variable increases, the other variable tends to increase as well; cold colors indicate negative correlations, meaning that as one variable increases, the other variable tends to decrease.

For instance, in Figure 2, the correlation coefficient between angle_1_0 and the tensile strength is 0.61, indicating a strong positive correlation, i.e., an increase in the angle value leads to an increase in the tensile strength. Conversely, the correlation coefficient between angle_1_45 and the tensile strength is −0.38, indicating a certain degree of negative correlation, i.e., an increase in the angle value leads to a decrease in the tensile strength. After data augmentation, as shown in Figure 3, although the correlation coefficients of most variables changed, the overall trend remained relatively stable. For example, the correlation coefficient between the cross-sectional area and strength decreased after augmentation but remained relatively high, indicating that the data augmentation method preserved the intrinsic relationships between variables, ensuring the model’s robustness and generalizability.

Overall, despite some changes in the variable correlations, most remained relatively stable, indicating that data augmentation did not significantly alter the characteristics of the data source. The analysis results suggest that the model built using this data augmentation method exhibits excellent performance, validating its effectiveness in predicting the mechanical properties of composite materials. This method not only successfully expanded the training set but also maintained the model’s robustness, significantly improving its generalizability. These data preprocessing steps provide a solid foundation for subsequent model training, ensuring the model can accurately capture patterns and effectively predict the mechanical properties of materials.

The research presented in this paper is primarily divided into three stages. The first stage is the data preparation phase, where initial data are obtained through experimental and numerical simulation methods. Subsequently, data augmentation techniques incorporating randomness were employed to expand the dataset, ensuring it could support the training requirements of machine learning models. The second stage involved model training and validation, where the models were optimized using grid search and cross-validation methods. The third stage comprised the comparison of predictions from different machine learning models, including a summary analysis of the model hyperparameters and performance validation results. By comparing the predictive outcomes of machine learning models with the findings from mechanistic studies, a multi-faceted understanding and interpretation of the performance of CFRP/metal composite materials were achieved, which offers new insights into the research in this field. Figure 6 summarizes the entire procedure.

## 4. Results and Discussion 

### 4.1. The Impact of Different Layup Sequences on the Tensile and Bending Properties of DP590/cfrp Composite Laminate

Table 2 presents the tensile and bending strengths of the DP590/CFRP composite laminates across 28 CFRP lamination sequences. The data were obtained using experimental methods under specific layup conditions (0°/90°/90°/90°/90°/0°). Subsequently, the validity of the established finite element model was verified using these experimental data. Utilizing the finite element model, the tensile and bending strengths of the DP590/CFRP composite laminates under various layup conditions were discussed.

The lamination sequence 2 (“0°/0°/0°/0°/0°/0°”) stands out by exhibiting the highest tensile strength at 819.97 MPa and the highest bending strength at 947.67 MPa. This superior performance is primarily attributed to the all-directional lamination approach, wherein fibers uniformly oriented at 0° ensure a more even stress distribution under both tensile and bending loads. This uniform stress distribution is pivotal in enhancing the laminate’s mechanical properties.

The analysis reveals that specific lamination sequences significantly influence the laminate’s mechanical behavior. For instance, sequences employing alternating fiber orientations, such as sequences 5 (“0°/90°/90°/90°/90°/0°”) and 6 (“0°/90°/0°/0°/90°/0°”), demonstrated improved tensile and bending performances due to better stress dispersion. Moreover, the inclusion of 45° fibers in sequences like 9 (“0°/0°/45°/−45°/0°/0°”) and 11 (“0°/45°/0°/0°/−45°/0°”) has been shown to optimize the tensile behavior in specific configurations further.

The findings underscore the critical role of the lamination sequence in optimizing the performance of fiber/metal composite materials. The all-directional lamination method, especially with uniform fiber orientation or strategic alternation, significantly contributes to the superior mechanical properties of composite laminates. These insights provide valuable guidance for the design and performance optimization of composite materials, highlighting the synergistic effect of fiber orientation in enhancing the laminate’s tensile and bending strengths.

Initially, the combination of finite element methods and experimental techniques was used to obtain the tensile and bending strengths of the DP590/CFRP composite laminates under 28 different layup sequences, as presented in Table 2. These values served as the raw data. To expand the dataset for machine learning purposes, a method involving the introduction of random bias was employed. Specifically, random bias coefficients were generated for certain fields within the dataset, such as the tensile and bending strengths, and these coefficients were applied to compute new field values, thereby synthesizing new data points. This randomization not only increased the quantity of the dataset but also enhanced its diversity, contributing to the model’s robustness and accuracy when dealing with real-world data. Ultimately, the newly generated dataset was sufficient to meet the demands of machine learning, allowing for further analysis and model training, as illustrated in Figure 7 and Figure 8. 

Figure 7 and Figure 8 show the fully expanded stretching and bending dataset after reasonable expansion. The term “Sample NO.” represents the serial number assigned to each experimental data point in our dataset. All experimental data are arranged in ascending order.

### 4.2. Predictive Results of Tensile Strength with Different Machine Learning Models

Figure 9 presents a comparative analysis of the tensile strength predictions made by various regression and machine learning models against the experimental values, highlighting their predictive accuracies and error distributions. Figure 9a illustrates the performance of the linear regression model, noting a maximum prediction error of −91.3843 MPa and a minimum of 0.0840 MPa, with 87 samples exceeding an absolute error of 20 MPa. The error spread predominantly ranges between −71 MPa and +59 MPa. Similarly, Figure 9b evaluates the ridge regression model, revealing a maximum error of −82.2506 MPa, a minimum of 0.0851 MPa, and 87 samples with errors above 20 MPa. Its error distribution extends from −69 MPa to +61 MPa. In the case of the lasso regression model, shown in Figure 9c, the errors stretch from −85.6135 MPa to −0.0688 MPa, with the same number of samples exhibiting significant errors. The error span is noted between −72 MPa and +60 MPa. Figure 9d assesses the K-nearest neighbor model, marking a notable improvement with a maximum error of 25.9403 MPa, a minimum of 0 MPa, and only four samples with errors beyond 20 MPa, showcasing a more contained error range from −16 MPa to +16 MPa.

The polynomial regression model’s performance is depicted in Figure 9e, where the maximum error reaches −131.8372 MPa and the minimum −0.0625 MPa, with 48 samples exceeding the 20 MPa error threshold. The error distribution is observed between −39 MPa and +26 MPa. The decision tree model evaluations in Figure 9f show a maximum error of −41.0732 MPa, a minimum of 0 MPa, and only two samples with significant errors, indicating an error distribution from −14 MPa to +14 MPa. Figure 9g explores the random forest model, presenting a maximum error of 35.4226 MPa, a minimum of 0.1779 MPa, and 18 samples with errors over 20 MPa. The error range is between −25 MPa and +14 MPa. Lastly, Figure 9h,i analyze the gradient boosting and XGBoost models, respectively. The former shows a maximum error of −21.7254 MPa and a minimum of 0.1260 MPa, with five samples exceeding the 20 MPa error benchmark and an error distribution from −19 MPa to +12 MPa. The latter demonstrates superior performance with no samples exceeding 20 MPa in error and a distribution range from −15 MPa to +13 MPa.

To assess the prediction errors, the XGBoost, gradient boosting, and K-nearest neighbors models exhibit a commendable reduction in maximum error, enhancing the reliability for practical use. The XGBoost (version number: 2.0.3.), decision tree, and K-nearest neighbors models also excel in stability, effectively managing high-error outliers. The error distribution of these models suggests a more accurate capture of the tensile strength characteristics, with XGBoost outperforming others in terms of a lower maximum error and stable predictive accuracy. Consequently, the XGBoost model is highlighted as the most effective, followed by the K-nearest neighbors and gradient boosting models for their satisfactory predictive capabilities.

Table 3 offers a detailed comparison of the performance and hyperparameter configurations of various machine learning models in predicting the tensile strength of CFRP/DP590. The evaluation encompasses the MAE, MSE, R^2^, MAPE, and the optimization of hyperparameters through methods such as grid search and cross-validation. The key insights from this comparison are summarized as follows:

XGBoost and gradient boosting are the top performers, which showcase exemplary precision with MAE values of 6.080 and 6.067, MSE scores of 56.15 and 59.86, R^2^ values at 0.996 for both, and MAPE at 1.08. Their hyperparameter configurations, finely tuned for optimal balance, underscore their superior performance.

Random forest exhibits solid results in the MAE, MSE, and R^2^ metrics, albeit slightly behind XGBoost and gradient boosting, hinting at the potential for further enhancement through hyperparameter optimization. K-nearest neighbors demonstrates commendable outcomes but reveals a higher MAPE, suggesting a possible sensitivity to noise in the dataset. Adjustments in its hyperparameters could mitigate this issue. Decision tree achieves satisfactory results with default parameters, though improvements in the MAE, MSE, and R^2^ could be achieved through tuning.

Polynomial regression underperforms, potentially due to overfitting, indicating a need to revisit the degree of polynomials used. Linear regression, lasso regression, and ridge regression are the least effective, with higher MAPEs and lower performance metrics, suggesting difficulty in capturing the data’s complexity.

In summary, XGBoost and gradient boosting are the most effective models, offering lower prediction errors and high explanatory power. Xing Liu et al. [58] have pointed out in their research that XGBoost, as a machine learning technique, is capable of providing accurate predictions for tabular datasets and possesses good predictive interpretability. The selection of a model should consider not only the performance metrics but also the complexity and interpretability to align with the requirements of practical applications.

### 4.3. Different Machine Learning Model Predictions for Bending Strength

Figure 10 offers a detailed comparison of the bending strength predictions from the nine distinct machine learning models against the experimental values, highlighting their predictive accuracy through specific metrics such as the maximum and minimum errors, the count of samples with significant errors (≥20 MPa), and the overall distribution of these errors. 

Linear regression showcased a broad error range with maximum and minimum errors of 367.8585 MPa and 0.1032 MPa, respectively, and a high number of samples (199) exceeding the 20 MPa error threshold, indicating significant variability in predictions. Ridge regression revealed a slightly improved accuracy with a maximum error of 369.6862 MPa and a minimum of −156.3737 MPa, yet it still had many samples (159) with significant errors. The lasso model presented a similar performance to ridge regression with a maximum error of 369.8354 MPa and 159 samples with errors above 20 MPa, indicating challenges in prediction accuracy. K-nearest neighbor (KNN) demonstrated a narrower error distribution from −32 MPa to +19 MPa and fewer samples (36) with significant errors, showcasing better model reliability. Polynomial regression exhibited the widest error range, with a maximum error of 381.2560 MPa, underscoring potential overfitting issues.

Decision tree regression showed improved precision with a maximum error of −41.5498 MPa and only 28 samples exceeding the 20 MPa error mark, indicating a more accurate prediction capability. The random forest model presented a balanced performance with a maximum error of −134.6730 MPa, and 65 samples had significant errors, suggesting a moderate level of prediction accuracy. Gradient boosting regression and XGBoost regression displayed concentrated error distributions and fewer samples with significant errors (29 and 34, respectively), indicating high consistency and reliability in their predictions. In conclusion, the decision tree, K-nearest neighbors, and random forest models were the top performers based on the smallest maximum errors, indicating their higher precision in bending strength predictions. The decision tree, gradient boosting, and XGBoost models had the fewest number of samples with significant errors, showcasing their reliability. The K-nearest neighbors, gradient boosting, and XGBoost models featured the most concentrated error distributions, highlighting their consistency and accuracy. These insights guide the selection of the optimal models for predicting the bending strength in materials science, emphasizing the importance of error minimization and predictive reliability.

Table 4 compares the prediction performance and hyperparameters of different machine learning models for the CFRP/DP590 tensile strength. The XGBoost and decision tree models exhibit excellent performance in terms of the MAE (9.66 and 9.91), MSE (136.35 and 145.46), R^2^ (0.98), and relatively low MAPE (1.156 and 1.191), respectively, demonstrating superior predictive accuracy and interpretability. The performance of these two models is relatively stable. During the hyperparameter tuning process, XGBoost used carefully adjusted hyperparameter settings, while the decision tree used default settings, indicating the exceptional performance of the decision tree on this problem. K-nearest neighbors and gradient boosting perform relatively well but are slightly inferior to XGBoost and decision tree. The hyperparameter settings for K-nearest neighbors {‘n_neighbors’: 11, ‘weights’: ‘distance’} reveal consideration for the distance weights of neighbors, while the settings for gradient boosting {‘learning_rate’: 0.33, ‘max_depth’: 6, ‘min_samples_leaf’: 1, ‘min_samples_split’: 4, ‘n_estimators’: 66} indicate a certain trade-off introduced in learning. The random forest performs quite ordinarily, showing a relatively poor performance, which might require further fine-tuning in the hyperparameters. Its settings {‘max_depth’: 19, ‘min_samples_leaf’: 1, ‘min_samples_split’: 2, ‘n_estimators’: 33} might lead to a rather complex model, calling for careful adjustments. Lasso regression, ridge regression, polynomial regression, and linear regression show the poorest performance in terms of the MAE, MSE, and R^2^, and have a relatively higher MAPE, indicating that linear models and polynomial regression might be inadequate to capture nonlinear relationships in the data. The hyperparameter settings for lasso regression and ridge regression are {‘alpha’: 0.18} and {‘alpha’: 5.25}, respectively, showing some balance achieved in the regularization process.

In summary, XGBoost and decision tree are the most outstanding models. K-nearest neighbors and gradient boosting are relatively good, while the performance of random forest is slightly worse. Relative to these, linear models and polynomial regression present the poorest performance. When choosing a model, the hyperparameters are closely related to the model’s performance and need careful adjustment to achieve the best prediction results.

In evaluating the predictive performance and hyperparameter settings of various machine learning models for the tensile strength of CFRP/DP590, distinct distinctions emerge. The XGBoost and decision tree models stand out for their exceptional accuracy, evidenced by their low mean absolute error (MAE) values of 9.66 and 9.91, mean squared error (MSE) values of 136.35 and 145.46, a consistent R^2^ of 0.98, and relatively minimal mean absolute percentage error (MAPE) of 1.156 and 1.191, respectively. Notably, the decision tree model achieves this high level of performance using default settings, highlighting its remarkable suitability for this application. In contrast, the XGBoost model requires finely tuned hyperparameters to achieve its optimal performance.

The K-nearest neighbors and gradient boosting models also demonstrate commendable performance but are slightly outperformed by XGBoost and decision tree. The hyperparameters for K-nearest neighbors (n_neighbors: 11, weights: ‘distance’) suggest a strategic emphasis on the influence of neighbor distances, whereas those for gradient boosting (learning_rate: 0.33, max_depth: 6, min_samples_leaf: 1, min_samples_split: 4, n_estimators: 66) reflect a deliberate balance between learning complexity and model efficiency.

The random forest model exhibits modest performance, indicating potential room for improvement through hyperparameter optimization. Its settings (max_depth: 19, min_samples_leaf: 1, min_samples_split: 2, n_estimators: 33) may contribute to its complexity, necessitating careful calibration to enhance its predictive capability.

Linear models, including lasso and ridge regression and polynomial and linear regression, perform poorly in comparison. Their higher MAE, MSE, R^2^ values, and MAPE suggest these models may not effectively capture the nonlinear dynamics present in the data. The hyperparameter settings for lasso (alpha: 0.18) and ridge regression (alpha: 5.25) attempt to strike a balance in regularization, yet their performance remains suboptimal.

In conclusion, the XGBoost and decision tree models are the top performers, offering both accuracy and interpretability. C. Furtado and colleagues have also reached similar conclusions, noting that XGBoost exhibits excellent performance in predicting the design of composite laminate panels [59]. K-nearest neighbors and gradient boosting are commendable but not at the same tier, while random forest requires further tuning for potential improvement. Linear models, including polynomial regression, prove less effective for this dataset, underscoring the importance of model selection and hyperparameter optimization in achieving precise predictive outcomes.

## 5. Conclusions

Through a comprehensive analysis of the DP590/CFRP composite laminate performance under various layup sequences, this study has identified key findings that substantially contribute to future material design and engineering applications:**Optimal Layup Sequences:** Layup sequence 2, employing an omnidirectional layup method, demonstrated superior mechanical properties with a tensile strength of 819.97 MPa and a bending strength of 947.67 MPa. Other sequences showing robust performance include layups 5, 6, 8, 9, 11, 15, 16, and 25, all exceeding 600 MPa in tensile strength, and sequences 5 through 12 for a bending strength above 900 MPa.**Machine Learning Model Performance:** Among the machine learning models evaluated, XGBoost and gradient boosting emerged as the top performers across multiple metrics, including maximum error, mean absolute error (MAE), mean squared error (MSE), and the coefficient of determination (R^2^). These models exhibited robustness and high interpretability, effectively capturing the complex relationships in the composite performance data.**Synergy Between Experimental and Numerical Approaches:** Integrating experimental data with numerical simulations and machine learning analysis has enriched our understanding of CFRP/steel composite materials. This holistic approach not only validates the finite element models but also enhances our insight into the material behavior under various conditions, demonstrating the complementary nature of these methodologies.

This study’s findings provide critical design references for optimizing the performance of fiber/metal composites and underline the effectiveness of advanced analytical models in predicting material behaviors.

## Figures and Tables

**Figure 1 polymers-16-01589-f001:**
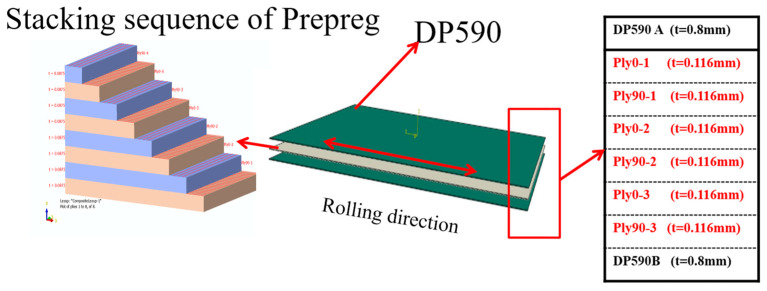
Schematic diagram of the laminated plate structure [47].

**Figure 2 polymers-16-01589-f002:**
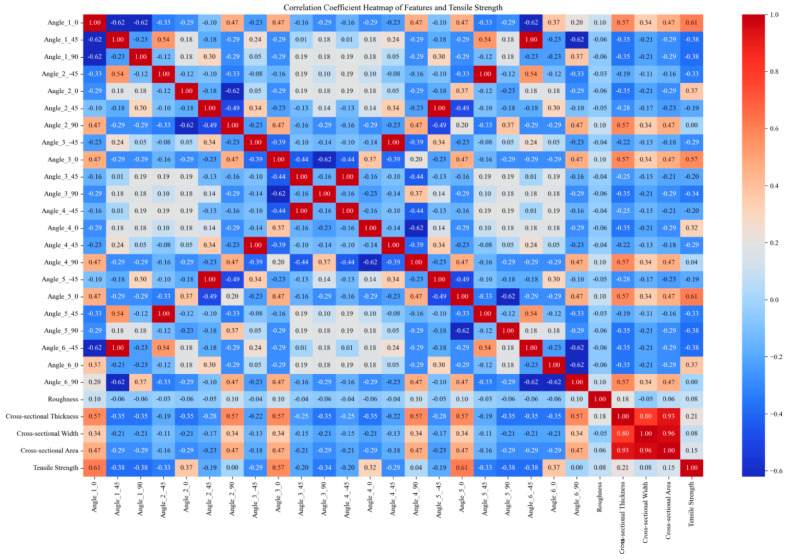
Heatmap of correlation coefficients before augmentation of tensile strength data.

**Figure 3 polymers-16-01589-f003:**
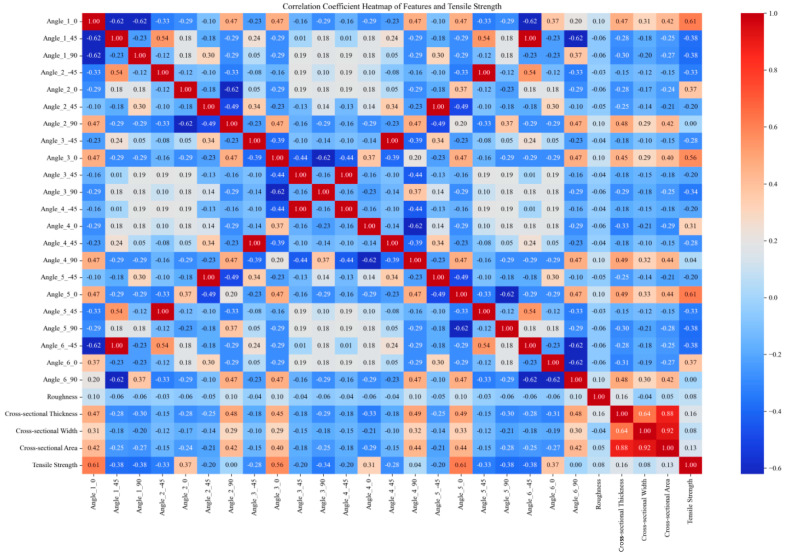
Heatmap of correlation coefficients after augmentation of tensile strength data.

**Figure 4 polymers-16-01589-f004:**
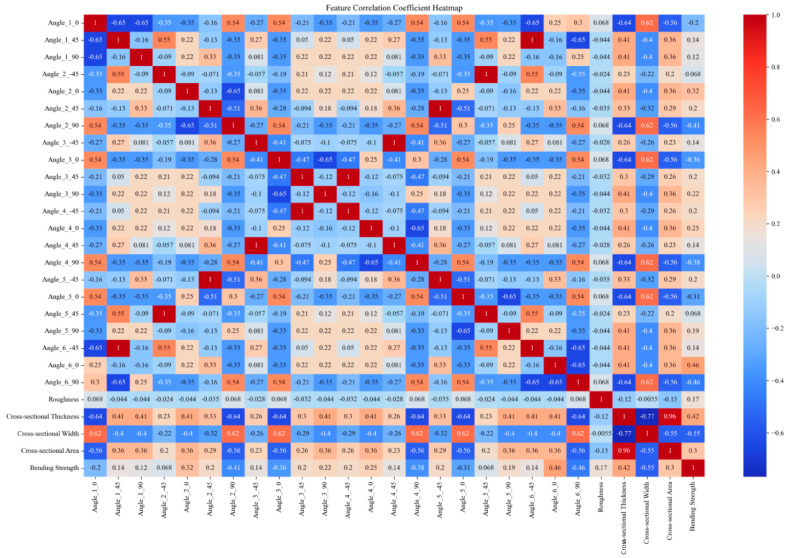
Heatmap of correlation coefficients before augmentation of bending strength data.

**Figure 5 polymers-16-01589-f005:**
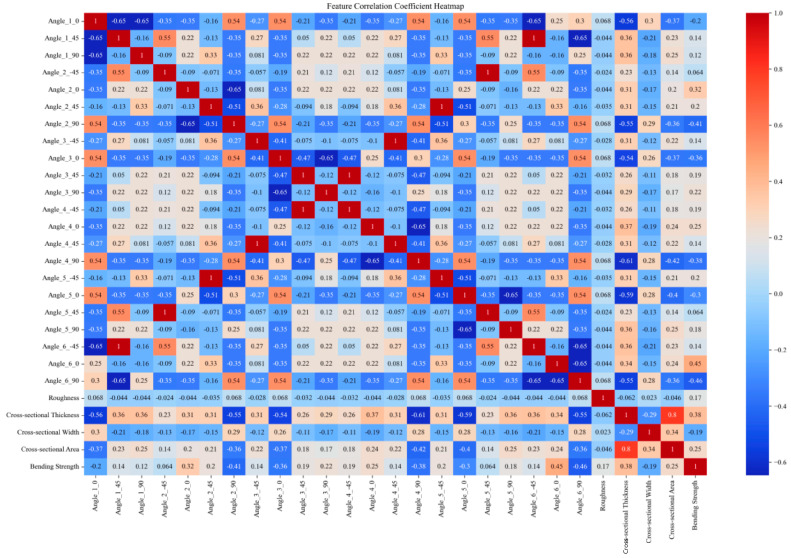
Heatmap of correlation coefficients after augmentation of bending strength data.

**Figure 6 polymers-16-01589-f006:**
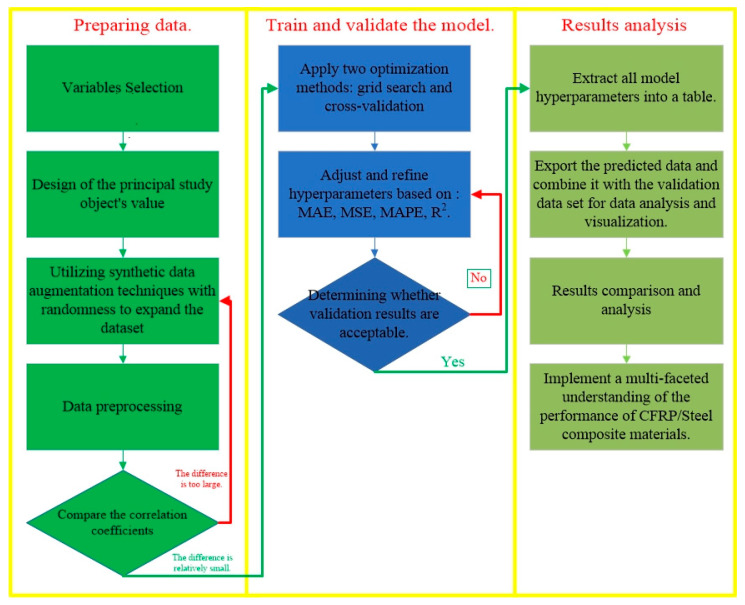
Three data processing stages. (Green represents the data preparation phase, symbolizing the inception and accumulation of data; blue is used for the model training and validation stage, emphasizing the stability and systematic nature of the process; light green is used for the prediction comparison and analysis stage, showing the natural transition from data preparation to application analysis).

**Figure 7 polymers-16-01589-f007:**
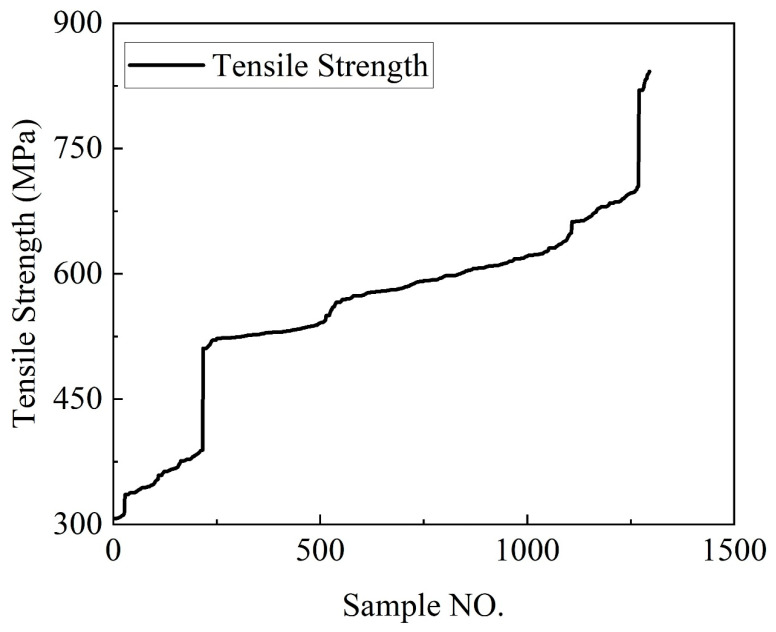
Stretching strength dataset.

**Figure 8 polymers-16-01589-f008:**
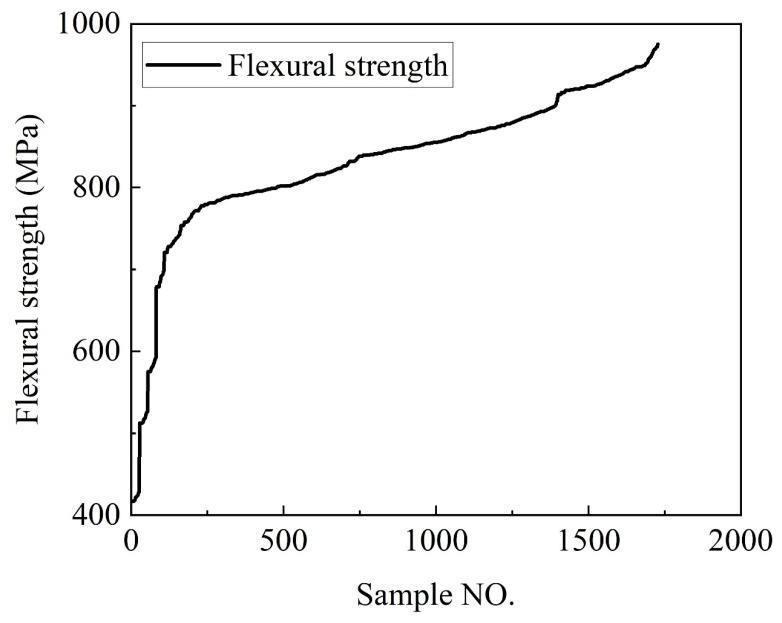
Bending strength dataset.

**Figure 9 polymers-16-01589-f009:**
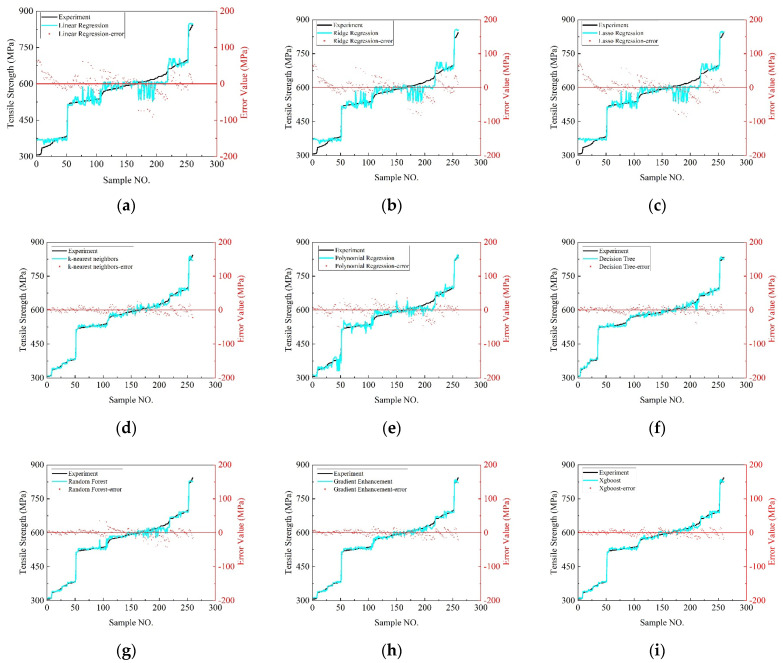
Comparison of tensile strength predictions by different machine learning models. (**a**) Linear regression. (**b**) Ridge regression. (**c**) Lasso regression. (**d**) K-nearest neighbors. (**e**) Polynomial regression. (**f**) Decision tree. (**g**) Random forest. (**h**) Gradient boosting. (**i**) Xgboost.

**Figure 10 polymers-16-01589-f010:**
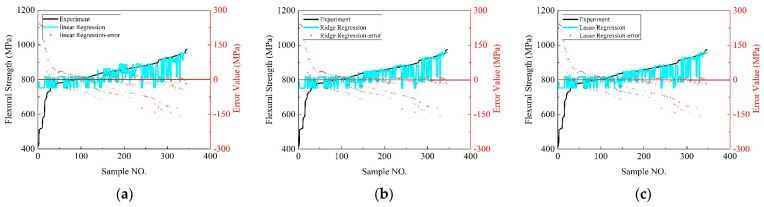
Comparison of bending strength prediction by different machine learning models. (**a**) Linear regression. (**b**) Ridge regression. (**c**) Lasso regression. (**d**) K−nearest neighbors. (**e**) Polynomial regression. (**f**) Decision tree. (**g**) Random forest. (**h**) Gradient boosting. (**i**) Xgboost.

**Table 1 polymers-16-01589-t001:** Layer design of CFRP laminate in tensile and bending models for DP590/CFRP composite panels.

Serial Number	Laying Sequence
1	0°/90°/0°/90°/0°/90°
2	0°/0°/0°/0°/0°/0°
3	90°/90°/90°/90°/90°/90°
4	45°/−45°/45°/−45°/45°/−45°
5	0°/90°/90°/90°/90°/0°
6	0°/90°/0°/0°/90°/0°
7	0°/90°/45°/−45°/90°/0°
8	0°/0°/90°/90°/0°/0°
9	0°/0°/45°/−45°/0°/0°
10	0°/45°/90°/90°/−45°/0°
11	0°/45°/0°/0°/−45°/0°
12	0°/45°/−45°/45°/−45°/0°
13	90°/90°/0°/0°/90°/90°
14	90°/90°/45°/−45°/90°/90°
15	90°/0°/90°/90°/0°/90°
16	90°/0°/0°/0°/0°/90°
17	90°/0°/45°/−45°/0°/90°
18	90°/45°/−45°/45°/−45°/90°
19	90°/45°/90°/90°/−45°/90°
20	90°/45°/0°/0°/−45°/90°
21	45°/90°/90°/90°/90°/−45°
22	45°/90°/0°/0°/90°/−45°
23	45°/90°/−45°/45°/90°/−45°
24	45°/0°/90°/90°/0°/−45°
25	45°/0°/0°/0°/0°/−45°
26	45°/0°/−45°/45°/0°/−45°
27	45°/−45°/90°/90°/45°/−45°
28	45°/−45°/0°/0°/45°/−45°

**Table 2 polymers-16-01589-t002:** Design of CFRP layer layout in the tension and bending models of DP590/CFRP composite laminate.

Serial Number	Laying Sequence	Tensile Strength (MPa)	Bending Strength (MPa)
1	0°/90°/0°/90°/0°/90°	578.96	892.76
2	0°/0°/0°/0°/0°/0°	819.97	947.67
3	90°/90°/90°/90°/90°/90°	307.10	831.92
4	45°/−45°/45°/−45°/45°/−45°	358.92	854.89
5	0°/90°/90°/90°/90°/0°	609.36	916.06
6	0°/90°/0°/0°/90°/0°	663.30	920.51
7	0°/90°/45°/−45°/90°/0°	530.09	920.68
8	0°/0°/90°/90°/0°/0°	662.96	941.62
9	0°/0°/45°/−45°/0°/0°	686.26	947.16
10	0°/45°/90°/90°/−45°/0°	524.35	918.80
11	0°/45°/0°/0°/−45°/0°	684.70	923.89
12	0°/45°/−45°/45°/−45°/0°	523.48	921.92
13	90°/90°/0°/0°/90°/90°	510.61	841.18
14	90°/90°/45°/−45°/90°/90°	338.09	840.21
15	90°/0°/90°/90°/0°/90°	606.61	867.77
16	90°/0°/0°/0°/0°/90°	662.26	872.63
17	90°/0°/45°/−45°/0°/90°	529.74	872.53
18	90°/45°/−45°/45°/−45°/90°	376.00	846.48
19	90°/45°/90°/90°/−45°/90°	344.31	839.51
20	90°/45°/0°/0°/−45°/90°	526.78	847.02
21	45°/90°/90°/90°/90°/−45°	335.86	839.65
22	45°/90°/0°/0°/90°/−45°	520.73	845.33
23	45°/90°/−45°/45°/90°/−45°	378.16	848.72
24	45°/0°/90°/90°/0°/−45°	523.23	870.43
25	45°/0°/0°/0°/0°/−45°	680.35	874.89
26	45°/0°/−45°/45°/0°/−45°	523.83	875.69
27	45°/−45°/90°/90°/45°/−45°	363.65	849.09
28	45°/−45°/0°/0°/45°/−45°	522.75	853.98

**Table 3 polymers-16-01589-t003:** Comparison table of prediction performance and hyperparameters of different machine learning models for CFRP/DP590 tensile strength.

Model	MAE	MSE	R^2^	MAPE	Hyperparameters
xgboost	6.080	56.15	0.996	1.08	{‘colsample_bytree’: 1, ‘learning_rate’: 0.5, ‘max_depth’: 5, ‘min_child_weight’: 1, ‘n_estimators’: 45, ‘subsample’: 1}
Gradient boosting	6.067	59.86	0.996	1.08	{‘learning_rate’: 0.19, ‘max_depth’: 5, ‘min_samples_leaf’: 1, ‘min_samples_split’: 7, ‘n_estimators’: 50}
Decision tree	6.469	66.20	0.994	1.13	default
K-nearest neighbors	6.81	73.01	0.995	1.19	{‘n_neighbors’: 13, ‘weights’: ‘distance’}
Random forest	7.35	107.53	0.992	1.28	{‘max_depth’: 9, ‘min_samples_leaf’: 4, ‘min_samples_split’: 2, ‘n_estimators’: 91}
Polynomial regression	8.82	152.54	0.989	1.54	{‘poly__degree’: 3}
Linear regression	18.91	682.98	0.950	3.65	default
Lasso regression	18.77	685.25	0.950	3.72	{‘alpha’: 0.41}
Ridge regression	19.19	689.05	0.950	3.74	{‘alpha’: 0.32}

**Table 4 polymers-16-01589-t004:** Comparison results of prediction performance and hyperparameters of different machine learning models for bending strength of CFRP/DP590.

Model	MAE	MSE	R^2^	MAPE	Hyperparameters
xgboost	9.66	136.35	0.983	1.156	{‘colsample_bytree’: 0.6, ‘learning_rate’: 0.66, ‘max_depth’: 5, ‘min_child_weight’: 1, ‘n_estimators’: 97, ‘subsample’: 1}
Decision tree	9.91	145.46	0.985	1.191	Default
K-nearest neighbors	11.61	341.60	0.957	1.379	{‘n_neighbors’: 11, ‘weights’: ‘distance’}
Gradient boosting	11.67	399.24	0.949	1.392	{‘learning_rate’: 0.33, ‘max_depth’: 6, ‘min_samples_leaf’: 1, ‘min_samples_split’: 4, ‘n_estimators’: 66}
Random forest	14.12	442.95	0.944	1.743	{‘max_depth’: 19, ‘min_samples_leaf’: 1, ‘min_samples_split’: 2, ‘n_estimators’: 33}
Lasso	40.72	4701.84	0.403	5.630	{‘alpha’: 0.18}
	40.80	4700.70	0.403	5.638	{‘alpha’: 5.25}
Ridge regression	43.35	4614.39	0.414	5.916	{‘poly__degree’: 2}
Polynomial regression	43.53	4799.88	0.390	5.938	Default

## Data Availability

The original contributions presented in the study are included in the article/Appendix A, further inquiries can be directed to the corresponding authors.

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
