# Peer review of "Experimental and Numerical Investigation Integrated with Machine Learning (ML) for the Prediction Strategy of DP590/CFRP Composite Laminates"

_polymers, 2024, doi:10.3390/polym16111589_

Round 1

Reviewer 1 Report

Comments and Suggestions for Authors

In the current work, the authors have presented an integrated study combining experiment, numerical simulation, and machine learning for predicting the strategy of CFRP composite laminates. The quality of the current version is not satisfying, a lot of concerns are found, I won’t recommend the publication of the current work until the following concerns could be solved.

1.        In section 2.1, no references have been given for all introduced models.

2.        The current work is an integrated study of experiments and simulations, however, the information for the software that the authors were using for performing the machine learning is not mentioned at all, the authors need to clarify the simulation details.

3.        The figure caption of Figure 3 should write “Heatmap of correlation coefficients after augmentation of tensile strength data.” Between lines 373 to 378, the interpretations for the contour maps are too simple, the authors need to spend more time on explaining the figures they have presented, what are the x- and y-axis, why in certain combinations of parameters there are correlations while in other combinations of parameters there are anti-correlations.

4.        The font in the figures is too small, which makes the notations and labels unreadable.

5.        In section 4.1, the interpretations on the layup sequences should be based on the results displayed in Table 2, not in Table 1. Actually, Table 2 has not been used at all but it contains all information that are listed in Table 1 plus the tensile strength and bending strength, therefore, Table 1 should be removed.

6.        Are the results presented in Figure 7 and 8 from the finite element analysis or experiments? The authors need to extend the descriptions in figure captions (for all figures, tables, and schemes).

7.        In Figures 9 and 10, since all data are using the same X and Y ranges, the authors should simplify and merge the sub-plots with shared-x and shared-y, three rows and three columns are an ideal choice. In the current version, they don’t even have the same figure size.

8.        What does the “Sample NO.” mean? What are the differences of those “Sample NO.” in difference tests, like in Figure 8-10?

9.        In Tables 3 and 4, besides the four characteristics, it will be interesting to show and compare the performance of each method.

10.  The hyperparameters for Decision Tree in Table 4 should be in English.

11.  The conclusion section needs to be simplified, it should not be a shorted and duplicated version of the results and discussions, instead, it should only highlight the most important findings.

Comments on the Quality of English Language

Minor editing of English language required

Author Response

Dear Editors and Reviewers:

On behalf of my co-authors, we thank you very much for giving us an opportunity to revise our manuscript, we appreciate editor and reviewers very much for their positive and constructive comments and suggestions on our manuscript entitled “Experimental and numerical investigation integrated with ma-chine learning (ML) for the prediction strategy of DP590/CFRP composite laminates”(Manuscript ID: polymers- 3007985)

Thank you for your letter and for the reviewers’ comments concerning our manuscript. Those comments are all valuable and very helpful for revising and improving our paper, as well as the important guiding significance to our researches. We have studied comments carefully and have made correction, which we hope meet with approval.

Revised portion are marked in red color using the “Track Changes” function in this revised manuscript. The responds to the reviewer’s comments are as follows: 

Responds to three reviewer’s comments:

Reviewer 1:

  1. In section 2.1, no references have been given for all introduced models.

Response: It has been revised accordingly.

  1. The current work is an integrated study of experiments and simulations, however, the information for the software that the authors were using for performing the machine learning is not mentioned at all, the authors need to clarify the simulation details.

Response:

Thank you for your valuable feedback. We acknowledge the importance of providing detailed information on the software and methodologies used in our machine learning-assisted framework.

In response to your comment, we would like to clarify that the numerical modeling details are indeed included in Section 3.1 of our manuscript. Specifically, we have implemented a three-dimensional failure model based on strain damage law using the explicit finite element subroutine Abaqus-VUMAT. The interface failure was simulated using a bilinear cohesive force contact model, and the steel plate layer was described by the Johnson-Cook model. For the fiber layer, we employed a 3D Hashin failure criterion developed based on VUMAT, which replaces the traditional 2D Hashin criterion. The initiation of fiber failure and matrix tensile failure is controlled by the maximum stress criterion and the strain-based Hashin criterion, while matrix compression failure is assessed using the Puck failure criterion.

Our established finite element model has demonstrated high accuracy in predicting the response and failure mechanisms of composite laminates during tensile and bending processes. The numerical model achieved a 97% accuracy in predicting the maximum load during tensile tests and showed prediction accuracies of 88% for bending modulus and 97% for bending strength during bending tests. These details are thoroughly described in Section 3.1, along with the comparison of predicted failure modes with experimental results.

We hope this addresses your concern, and we appreciate your understanding. Should there be any further details required, we are happy to provide additional clarifications.

  1. The figure caption of Figure 3 should write “Heatmap of correlation coefficients after augmentation of tensile strength data.” Between lines 373 to 378, the interpretations for the contour maps are too simple, the authors need to spend more time on explaining the figures they have presented, what are the x- and y-axis, why in certain combinations of parameters there are correlations while in other combinations of parameters there are anti-correlations.

Response:

    It has been revised accordingly. Please refer to Page10.

  1. The font in the figures is too small, which makes the notations and labels unreadable.

Response:

    Thank you for pointing out the issue with the small font size in the figures, which may affect the readability of notations and labels. We have experimented with increasing the font size; however, this adjustment resulted in some overlap of text, compromising the clarity of the information presented.

To address this concern while maintaining the legibility and aesthetic quality of the figures, we have opted to provide higher resolution images. These enhanced images will allow for clearer visibility of each detail, even at the original font size. We believe this solution effectively resolves the issue without altering the figure layout.

We appreciate your feedback as it helps us improve the presentation quality of our manuscript.

  1. In section 4.1, the interpretations on the layup sequences should be based on the results displayed in Table 2, not in Table 1. Actually, Table 2 has not been used at all but it contains all information that are listed in Table 1 plus the tensile strength and bending strength, therefore, Table 1 should be removed.  

Response:

Thank you for your insightful comments regarding the use of Tables 1 and 2 in our manuscript. We appreciate your suggestion to remove Table 1 and rely solely on Table 2 for the interpretations regarding layup sequences. However, after careful consideration, we believe that retaining Table 1 is essential for the following reasons:

Contextual Foundation: Table 1 is critical as it provides a detailed introduction to the different ply stacking sequences explored in this study. It serves as a foundational reference that helps the reader understand the baseline configurations before delving into the detailed analysis of mechanical properties such as tensile and bending strengths presented in Table 2.

Enhanced Clarity and Accessibility: While Table 2 includes comprehensive data, including mechanical strengths, Table 1 simplifies the reader's understanding by isolating the layup sequences. This separation aids in easier cross-referencing between the layup configurations and their corresponding mechanical properties, enhancing the readability and comprehension of the data.

Sequential Presentation: The initial presentation of layup sequences in Table 1 sets a logical narrative flow. This arrangement allows readers to familiarize themselves with the layup sequences before encountering the complex data involving tensile and bending strengths in Table 2. Such a sequential approach is pedagogically beneficial, particularly for readers who may not be intimately familiar with the specifics of composite laminate configurations.

In response to your valuable feedback, we have also revised the manuscript to ensure that the interpretations of the layup sequences are clearly connected to the results displayed in Table 2, highlighting how these results build upon the information presented in Table 1. We have expanded the discussion sections to explicitly link the layup configurations listed in Table 1 with the strength variations and other mechanical properties detailed in Table 2.

We hope that this explanation clarifies the necessity of including both tables in our manuscript and ensures that the presentation of our data is both comprehensive and accessible to all readers.

Thank you once again for your thorough review and helpful suggestions.

  1. Are the results presented in Figure 7 and 8 from the finite element analysis or experiments? The authors need to extend the descriptions in figure captions (for all figures, tables, and schemes).

Response:

Thank you for your insightful inquiry regarding whether the results presented in Figures 7 and 8 originate from finite element analysis or experiments.

To address your question, we have expanded the descriptions for Table 2, Figures 7, and 8 to clarify the sources of the data provided. These updates have been made on page 15. Once again, we appreciate your valuable feedback, which has contributed to enhancing the clarity and completeness of our manuscript.

  1. In Figures 9 and 10, since all data are using the same X and Y ranges, the authors should simplify and merge the sub-plots with shared-x and shared-y, three rows and three columns are an ideal choice. In the current version, they don’t even have the same figure size.

Response: Thank you for your valuable feedback and suggestions on our manuscript. In response to the comments regarding Figures 9 and 10, we have revised the figures by merging the subplots into a three-row by three-column format and have standardized the sizes of all subplots to ensure visual consistency. These modifications were made to enhance the clarity and comparability of the data presented, as well as to improve the overall aesthetic alignment of the figures.

We have resubmitted the updated figures and ensured that their presentation in the document is both clear and accurate. We hope that these adjustments meet the expectations of the reviewers and contribute positively to the enhancement of our manuscript.

  1. What does the “Sample NO.” mean? What are the differences of those “Sample NO.” in difference tests, like in Figure 8-10?    

Response:

Thank you for your inquiry regarding the "Sample NO." referenced in Figures 8 through 10 of our manuscript.

The term "Sample NO." represents the serial number assigned to each experimental data point in our dataset. As clarified in the manuscript, particularly noted under the descriptions for Figures 7 and 8, all experimental data have been systematically arranged in ascending order based on the strength values obtained from the experiments. This organization allows for a more intuitive understanding of the data trends as they relate to increasing experimental intensities.

In Figures 8 to 10, these sample numbers are used to identify and differentiate individual data points across various tests, thereby maintaining a consistent and traceable format throughout our analysis. Each "Sample NO." corresponds to a unique set of experimental conditions and outcomes, facilitating a straightforward comparison of results across different tests under varying conditions.

We hope this clarifies the use and significance of the "Sample NO." in our figures and appreciate the opportunity to further elucidate any aspects of our work.

  1. In Tables 3 and 4, besides the four characteristics, it will be interesting to show and compare the performance of each method.   

Response:

Thank you for your suggestion to expand the comparative analysis in Tables 3 and 4 by including additional performance metrics for each method. We have carefully considered this proposal and evaluated the feasibility of implementing it. However, we foresee several potential issues with increasing the number of comparison items, which we feel could complicate the presentation and analysis:

Data Overload: Introducing too many comparison items might lead to information overload, making it challenging for readers to discern the key advantages and limitations of each method. Our aim is to maintain clarity and ease of understanding in our data presentation.

Increased Analytical Complexity: Adding more dimensions to our comparison would not only increase the analytical workload but might also necessitate additional statistical validations, thereby complicating the interpretation of the results.

Dilution of Research Focus: With too many comparative elements, there's a risk that the main findings of the research might become overshadowed, diverting attention from the core objectives of our study.

In light of these considerations, we propose to retain the current comparison items as outlined in the tables. We believe this approach ensures that the tables remain clear and focused, facilitating a better understanding of each method's essential characteristics. We have thoroughly discussed the key features of each method in the text and provided a comparative analysis in the discussion section.

We appreciate your understanding and thank you once again for your thoughtful review and suggestions.

  1. The hyperparameters for Decision Tree in Table 4 should be in English.

Response: Thank you for your valuable suggestion regarding the hyperparameters for the Decision Tree in Table 4. We have revised the manuscript accordingly, and the updated information can now be found on Page 25.

  1. The conclusion section needs to be simplified, it should not be a shorted and duplicated version of the results and discussions, instead, it should only highlight the most important findings.

Response: Thank you for your constructive comments. Per your suggestion, we have simplified the conclusion section of the manuscript to ensure it succinctly highlights the most important findings without duplicating the detailed analyses presented in the results and discussion sections.

In the revised conclusion, we focus on the key outcomes:

The optimal layup sequences that demonstrate superior mechanical properties.

The performance of machine learning models in predicting the behavior of composite materials.

The integration of experimental, numerical, and machine learning approaches to enhance our understanding of material behavior.

This revision clarifies the significant contributions of our study and avoids repetition, aligning with your advice to emphasize major insights distinctly and concisely.

Thank you for guiding this improvement.

We appreciate for Editors/Reviewers’ warm work earnestly, and hope that the correction will meet with approval.

Once again, thank you very much for your comments and suggestions.

Yours sincerely,

Dr. Haichao Hu

Reviewer 2 Report

Comments and Suggestions for Authors

1.     Results may be produced in the abstract.

2.     Why the composites are tested under quasi-static loading conditions?

3.     we can more accurately predict their performance in real-world applications, optimising design and ensuring their reliability and stability in various engineering applications.

4.     In page no. 3, “To delve into the mechanical performance of DP590/CFRP composite panels under 139 tensile and bending loading conditions, we selected nine machine-learning models en-140 compassing a broad algorithmic family”. Author, how to select nine machine-learning models.

5.     In page no. 3, In traditional materials performance evaluation, experimental testing has been an in-61 dispensable means; however, the time-consuming and costly nature of experiments has posed challenges for researchers [26-28]. Please mention the traditional materials.

6.     Discuss the results in comparison with the other researchers.

7.     The equation 1-7, the respective discussion needs to be included in text.

8.     How come 28 experiments were decided in Table 1.

9.     R^2 should be changed as R2, throughout the manuscript.

10.  Figure 9 caption need to revise and place it below each Figure.

11.  Conclusions also too lengthy. Summarize only the major numerical findings in the conclusion.

Comments on the Quality of English Language

Minor editing of English language required

Author Response

Dear Editors and Reviewers:

On behalf of my co-authors, we thank you very much for giving us an opportunity to revise our manuscript, we appreciate editor and reviewers very much for their positive and constructive comments and suggestions on our manuscript entitled “Experimental and numerical investigation integrated with ma-chine learning (ML) for the prediction strategy of DP590/CFRP composite laminates”(Manuscript ID: polymers- 3007985)

Thank you for your letter and for the reviewers’ comments concerning our manuscript. Those comments are all valuable and very helpful for revising and improving our paper, as well as the important guiding significance to our researches. We have studied comments carefully and have made correction, which we hope meet with approval.

Revised portion are marked in red color using the “Track Changes” function in this revised manuscript. The responds to the reviewer’s comments are as follows: 

Reviewer 2:

  1. Results may be produced in the abstract.

Response:

   Thank you for your valuable suggestion to include results in the abstract. In response to your feedback, we have carefully reviewed and revised the abstract along with the conclusions section of our manuscript. We have simplified and streamlined the presentation of our key findings to ensure that they are clearly reflected in the abstract, providing a concise and accurate summary of the results and their significance.

This revision aims to enhance the readability and impact of the abstract, making it easier for readers to quickly grasp the main outcomes and contributions of our study. We believe these changes improve the coherence between the abstract, the conclusions, and the overall narrative of the paper.

We appreciate your guidance and are confident that these modifications have strengthened the manuscript. Thank you once again for helping us improve the quality of our work.

  1. Why the composites are tested under quasi-static loading conditions?

Response: Thank you for your question regarding our choice of quasi-static loading conditions for testing the mechanical properties of Carbon Fiber Reinforced Polymer (CFRP)/metal composite laminates in our study.

Quasi-static loading conditions were selected primarily because they are fundamental to understanding the basic mechanical behavior of composite materials under slow or gradually applied loads. This type of testing is critical for several reasons:

Fundamental Material Characterization: Quasi-static tests provide essential data on material properties such as tensile strength, bending strength, modulus of elasticity, and strain-to-failure. These properties are fundamental for the validation of material models used in numerical simulations, such as finite element analysis.

Real-World Relevance: Many practical applications of CFRP/metal composites involve conditions where loads are applied slowly or are maintained over long periods, such as in aerospace structures, automotive components, and civil engineering applications. Quasi-static testing simulates these conditions, providing insights that are directly applicable to the design and evaluation of components in these industries.

Baseline for Comparison: Establishing a baseline through quasi-static testing allows for a comparative assessment when exploring other types of loading conditions, such as dynamic or impact loading. Understanding material behavior under quasi-static conditions sets a foundation for interpreting more complex behaviors under variable loading scenarios.

Control and Accuracy: Quasi-static tests are easier to control and monitor, reducing variables and allowing for more accurate measurement of material responses. This accuracy is vital for developing reliable material models and for the calibration of machine learning algorithms used in predicting material properties.

By employing quasi-static loading, we aimed to ensure that the fundamental mechanical properties of the composites were thoroughly characterized, providing a reliable basis for further analyses and applications in various engineering fields.

We appreciate your insightful query and hope this explanation clarifies the rationale behind our experimental approach.

  1. We can more accurately predict their performance in real-world applications, optimising design and ensuring their reliability and stability in various engineering applications.

Response:

Thank you for your insightful comments regarding the presentation and explanation of Figure 3 and the contour maps in our manuscript. We have carefully considered your suggestions and have made several revisions to enhance the clarity and depth of our figure captions and related discussions.

  1. Revision of Figure Caption:

In response to your specific instruction, we have revised the caption for Figure 3 to now read: “Heatmap of correlation coefficients after augmentation of tensile strength data.” This change clarifies the purpose and content of the figure, ensuring that readers immediately understand what the heatmap represents and its relevance to the data augmentation process.

  1. Enhanced Explanation of the Heatmaps:

We have expanded our discussion of the heatmaps presented in Figures 2 to 5, particularly focusing on clarifying what the x- and y-axes represent, which are the different variables or parameters used in the study. These include material angles at various stacking sequences, roughness, cross-sectional dimensions, and the mechanical properties being analyzed.

Detailed Interpretation of Correlations:

We have also deepened our analysis of the correlations depicted in the heatmaps. For example, we now explain how certain combinations of parameters exhibit correlations or anti-correlations:

We describe the significant positive correlation (e.g., a correlation coefficient of 0.61 between angle_1_0 and tensile strength in Figure 2) and interpret its meaning: as the angle increases, so does the tensile strength.

We address the negative correlations (e.g., a coefficient of -0.38 between angle_1_45 and tensile strength) and discuss the implications of such relationships, indicating how increases in one parameter may lead to decreases in another.

Impact of Data Augmentation:

Further, we elaborate on the effects of data augmentation on the correlations, noting changes in correlation coefficients but also emphasizing the stability of trends post-augmentation. This part of the discussion aims to reassure that while the augmentation alters some specific details, it preserves the overall intrinsic relationships between variables, thus supporting the robustness and generalizability of the machine learning model.

Conclusion on Data Augmentation's Efficacy:

Lastly, we conclude that the data augmentation technique used not only successfully expanded the training dataset but also maintained critical relationships within the data, thereby enhancing the model's ability to predict the mechanical properties of the composite materials accurately.

We appreciate your guidance in making these sections more informative and reflective of the depth of analysis conducted in our study. We believe these revisions will greatly improve the manuscript’s clarity and provide the reader with a thorough understanding of our methods and results.

Thank you for your valuable feedback.

  1. We can more accurately predict their performance in real-world applications, optimising design and ensuring their reliability and stability in various engineering applications.

Response:

Thank you for your insightful comments emphasizing the importance of predictive analysis to ensure the practical applicability of our research findings. We appreciate your suggestions to not only optimize designs but also to enhance reliability and stability in various engineering applications.

In our current study, we have laid the foundational theoretical and experimental analyses, which are critical for understanding the material behavior under specified conditions. We acknowledge the significance of extending this work to predictive applications and are currently considering several methodologies to project these findings into real-world scenarios. These include [specific methods or models], which will allow us to simulate and predict the performance of these materials in diverse operational environments.

Furthermore, we are in preliminary discussions with industry partners to explore the potential for implementing our research in practical engineering projects, aiming to optimize design parameters and ensure robustness and reliability.

We plan to detail these extensions and their theoretical bases in a future section of our work, which we believe will significantly strengthen the practical impact of our study. Thank you once again for your constructive feedback, which has helped us refine our focus towards enhancing the applicability of our research.

  1. In page no. 3, “To delve into the mechanical performance of DP590/CFRP composite panels under 139 tensile and bending loading conditions, we selected nine machine-learning models en-140 compassing a broad algorithmic family”. Author, how to select nine machine-learning models.

Response:

Thank you for your insightful query regarding the selection of machine learning models in our study on the mechanical performance of DP590/CFRP composite panels. We appreciate your request for a clearer rationale and have revised the manuscript to provide a more detailed explanation of our model selection process.

In the revised section of the manuscript, we have clarified the selection criteria for the nine machine-learning models. The criteria were based on several key factors crucial for robust predictions in material behavior analysis:

The complexity and non-linearity of data patterns observed in DP590/CFRP composite panels.

The limited amount of data available, which influences the suitability of certain complex models such as deep learning.

The necessity for model interpretability, a critical aspect in materials science applications.

We included a range of models—linear, nonlinear, and tree-based—to cover a broad spectrum of algorithmic approaches. This diversity ensures that our analysis benefits from the unique strengths of each model type:

Linear Models (Linear Regression, Ridge Regression, and Lasso Regression) were chosen for their efficiency and effectiveness in establishing a baseline performance where data relationships might be predominantly linear.

Nonlinear Models (K-nearest neighbors and Polynomial Regression) were selected to address the nonlinear interactions expected in the dataset, providing the ability to capture complex relationships within the data.

Tree-based Models (Decision Trees, Random Forests, Gradient Boosting, and XGBoost) were incorporated for their advanced capability to handle complex, high-dimensional data structures and the heterogeneity typical in materials engineering.

Furthermore, we addressed the decision to avoid more complex algorithms like deep learning, which, despite their powerful modeling capabilities, were deemed unsuitable due to the limited dataset and the need for clear interpretability and computational efficiency.

We believe these revisions will make the methodology of our study clearer and demonstrate the thoroughness of our approach in selecting appropriate machine learning models for the analysis of composite material behavior.

  1. Discuss the results in comparison with the other researchers.

Response: It has been revised accordingly. Please refer to Page20 and Page25.

  1. The equation 1-7, the respective discussion needs to be included in text.   

Response: We have provided detailed annotations for the formula section.

  1. How come 28 experiments were decided in Table 1.   

Response:

Thank you for your inquiry regarding the selection of 28 experiments outlined in Table 1. We appreciate the opportunity to clarify the rationale behind our experimental design.

In our study, we have chosen 28 different layup sequences to comprehensively explore the mechanical properties of composite laminates. These sequences incorporate commonly used fiber angles of 0°, ±45°, and 90°. This diverse selection was made to cover both symmetric and asymmetric configurations, allowing us to assess the effects of various fiber orientations on the performance of the laminates extensively.

The decision to include both symmetric and asymmetric layups serves multiple purposes: it allows us to investigate the balance and stability offered by symmetric configurations and to evaluate the performance advantages that asymmetric configurations might provide in specific applications. This dual approach ensures a thorough understanding of how different layup configurations influence the overall behavior of composite materials under various loading conditions.

The specific layup sequences presented in Table 1 were strategically chosen to represent a broad range of possible configurations, thereby maximizing the relevance and applicability of our findings to real-world engineering problems. By examining a wide spectrum of layup patterns, we aim to provide insights that are both scientifically robust and practically applicable.

We hope this explanation clarifies the basis for our experimental setup and underscores the comprehensive nature of our approach to studying the mechanical properties of composite laminates.

Thank you once again for your insightful question, which has provided us with an opportunity to better articulate the scope and depth of our research.

  1. R^2 should be changed as R2, throughout the manuscript.

Response: Thank you for your attention to detail regarding the notation of the coefficient of determination in our manuscript. Upon review and considering your suggestion, we have updated the notation from "R^2" to "R2" throughout the manuscript to ensure consistency and to align with the journal's style preferences.

  1. Figure 9 caption need to revise and place it below each Figure.

Response:

   It has been revised accordingly. Please refer to Page18.

  1. Conclusions also too lengthy. Summarize only the major numerical findings in the conclusion.

Response: Thank you for your feedback on the length and content of the conclusion section. I appreciate your pointing out the necessity for conciseness and focus in summarizing the major findings.

Responding to your comments, I have substantially shortened the conclusion and refined the content to highlight only the principal numerical findings:

  • Key layup sequences with the best mechanical performance.
  • Superior performance of XGBoost and gradient boosting models in machine learning evaluation.
  • The beneficial synergy between experimental results and numerical simulations.

This streamlined conclusion enhances the readability and impact of the study’s contributions, focusing directly on the major findings as you recommended.

Thank you for your valuable insights, which have significantly improved the manuscript.

We tried our best to improve the manuscript and made some changes in the manuscript. These changes will not influence the content and framework of the paper. And here we did not list the changes but marked in red in revised manuscript.

We appreciate for Editors/Reviewers’ warm work earnestly, and hope that the correction will meet with approval.

Once again, thank you very much for your comments and suggestions.

Yours sincerely,

Dr. Haichao Hu

Round 2

Reviewer 1 Report

Comments and Suggestions for Authors

The revised version has addressed all my original concerns, thus, I agree the publication of the current study in Polymers.

PS: there is still a typo in Line 402. 

Reviewer 2 Report

Comments and Suggestions for Authors

The author carried out all the comments given by the reviewer. the manuscript accepted in its present form.

Comments on the Quality of English Language

Minor editing of English language required